# Fully Interpretable Deep Learning Model Using IR Thermal Images for Possible Breast Cancer Cases

**DOI:** 10.3390/biomimetics9100609

**Published:** 2024-10-09

**Authors:** Yerken Mirasbekov, Nurduman Aidossov, Aigerim Mashekova, Vasilios Zarikas, Yong Zhao, Eddie Yin Kwee Ng, Anna Midlenko

**Affiliations:** 1School of Engineering and Digital Sciences, Nazarbayev University, Astana 010000, Kazakhstannurduman.aidossov@nu.edu.kz (N.A.); yong.zhao@nu.edu.kz (Y.Z.); 2Department of Mathematics, University of Thessaly, GR-35100 Lamia, Greece; vzarikas@uth.gr; 3Mathematical Sciences Research Laboratory (MSRL), GR-35100 Lamia, Greece; 4School of Mechanical and Aerospace Engineering, Nanyang Technological University, Singapore 639798, Singapore; mykng@ntu.edu.sg; 5School of Medicine, Nazarbayev University, Astana 010000, Kazakhstan; anna.midlenko@nu.edu.kz

**Keywords:** breast cancer, Bayesian networks, convolutional neural networks, explainable artificial intelligence, machine learning, thermography

## Abstract

Breast cancer remains a global health problem requiring effective diagnostic methods for early detection, in order to achieve the World Health Organization’s ultimate goal of breast self-examination. A literature review indicates the urgency of improving diagnostic methods and identifies thermography as a promising, cost-effective, non-invasive, adjunctive, and complementary detection method. This research explores the potential of using machine learning techniques, specifically Bayesian networks combined with convolutional neural networks, to improve possible breast cancer diagnosis at early stages. Explainable artificial intelligence aims to clarify the reasoning behind any output of artificial neural network-based models. The proposed integration adds interpretability of the diagnosis, which is particularly significant for a medical diagnosis. We constructed two diagnostic expert models: Model A and Model B. In this research, Model A, combining thermal images after the explainable artificial intelligence process together with medical records, achieved an accuracy of 84.07%, while model B, which also includes a convolutional neural network prediction, achieved an accuracy of 90.93%. These results demonstrate the potential of explainable artificial intelligence to improve possible breast cancer diagnosis, with very high accuracy.

## 1. Introduction

Cancer is characterized by irregular cell division in the body, which accelerates the formation of tumors [1]. It represents a major health problem worldwide, being the second leading cause of death in the world and causing approximately ten million deaths each year [2]. Over the past 10 years, cancer mortality rates have increased by almost 30%. Breast and lung cancers are among the most common types in the world. For women in particular, breast cancer poses a significant health risk because, according to statistics, one in eight women is at risk of developing it during her lifetime [3]. In 2020, the World Health Organization (WHO) stated that breast cancer was the most commonly diagnosed cancer, with 2.3 million cases reported, resulting in the loss of 0.7 million lives [4], highlighting the critical nature of the problem. It is also expected that by 2025, the incidence of breast cancer will increase by 1.3 times among women over 30 years of age [5]. In some Asian countries, problems such as access to treatment, misdiagnosis, and long waiting periods in healthcare facilities hinder timely diagnosis and treatment of breast cancer [6]. Unfortunately, effective control of breast tumor growth remains a challenge [7].

According to the WHO, widespread adoption of breast self-examination (BSE) can be an effective means of combating breast cancer [8]; this involves regularly examining breasts for lumpiness or tissue thickness [9]. Timely detection is imperative for precise diagnosis, given that breast cancer contributes to approximately 15% of cancer-related fatalities [10,11,12,13,14]. With recent advances in technology, artificial intelligence (AI) may be a potentially useful tool for BSE. AI may offer a safer alternative, beneficial for women who are at increased risk of breast cancer or have a family history of breast cancer [10].

Routine voluntary screening assumes pivotal importance in curtailing breast cancer mortality rates; however, the screening process must be economically viable, safe, and patient-friendly. Although an array of imaging modalities, encompassing mammography, ultrasound, magnetic resonance imaging (MRI), and thermography, are employed for diagnostic purposes, each is beset by constraints such as diminished sensitivity, exposure to radiation, discomfort, financial exigency, and a predisposition toward yielding false positive outcomes [15]. Consequently, researchers’ increased attention has been aimed at improving and facilitating methods for early detection of cancer [16].

The most widely used diagnostic method is mammography, which involves imaging and uses X-rays to detect abnormalities in the mammary glands [17,18,19]. Ultrasound imaging uses acoustic waves and acts as an additional diagnostic tool, but its accuracy depends on the accuracy of the equipment, operator skill, and interpretation experience [15]. MRI is a costly method for breast cancer detection; in addition, women living in remote areas have limited access to MRI.

Recent studies have demonstrated the effectiveness of thermography as an initial screening method for detecting breast cancer. It is a safe, cost-effective, and non-invasive approach that can visualize breast surface temperature distribution and detect temperature anomalies or uneven patterns, which may indicate early-stage tumors within the breast [20]. As a valuable supplementary technique, thermography can identify subtle temperature changes that signal developing abnormalities, providing a painless and radiation-free option suitable for women of all ages, particularly those with dense breast tissue or implants [21,22,23,24,25,26,27]. Additionally, infrared (IR) cameras used for thermography are relatively inexpensive, fast, and simple, making them ideal for rapid breast cancer screening [23]. However, one of the main limitations of thermography is its heavy reliance on the physician’s expertise and experience, increasing the likelihood of human error.

Table 1 presents a comparison of the imaging techniques for breast cancer discussed above.

Considering the limitations of thermography and the advanced development of artificial intelligence techniques, integrating AI with thermography holds the potential to significantly enhance early-stage breast cancer detection.

This research aims to harness interpretable ML methodologies to enhance early detection of breast cancer, with objectives centered on improving accuracy and mitigating risks associated with misdiagnosis and radiation exposure. Leveraging Explainable Artificial Intelligence, (XAI), techniques, particularly focusing on thermography, allows efficient processing of large datasets to identify patterns that significantly enhance algorithm effectiveness as dataset size increases [24]. To streamline radiologists’ workload and cut costs, a proposed novel computer detection system is suggested for lesion classification and accurate identification. ML widely and effectively utilized across various domains finds suitability in processing and predicting breast cancer data [25,26,27]. Integration of Convolutional Neural Networks (CNNs) with Bayesian networks (BNs) presents advantages such as effective performance with large datasets, minimized error rates, and most importantly, interpretability, which is very important for a physician.

Biomimetic principles in image processing for medical image classification, detection, localization, and segmentation for medical diagnostics refer to the application of concepts and mechanisms observed in biological systems—especially in living organisms—into the design and implementation of artificial image processing techniques and diagnostic procedures [28,29]. These biomimetic approaches seek to enhance the capabilities of artificial systems by mimicking the highly efficient and robust mechanisms evolved by nature, particularly the human visual system and other biological processes. The present study complies fully with all the elements of this strategy. We propose a new method that detects the presence of tumors based on breast thermal images using biomimetic convolutional neural networks (a type of artificial neural network), Bayesian networks and XAI techniques that segment and spot the critical parts of the image that causally influence the diagnosis, exactly like the human brain does. Furthermore, at an even deeper level, our method suggests the concepts/factors that are responsible for the diagnosis, like the human brain of a physician that classifies and distinguishes particular important features of the image. In the following paragraphs, we elaborate on this issue.

The relationship between the brain’s neural networks and artificial neural networks (ANNs) forms the foundation of much of modern artificial intelligence, especially in fields like machine learning and deep learning. While ANNs were initially inspired by the workings of the brain, they have evolved in ways that are both similar and distinct from biological systems. In ANNs, the basic unit is a node, which mimics the function of a biological neuron. A node receives inputs, processes them, and produces an output. Just like a biological neuron, the output depends on the inputs and their weights (analogous to synaptic strength). The connections between artificial neurons are represented by weights, which determine the strength and influence of one neuron’s signal on another. Learning in ANNs involves adjusting these weights to improve the network’s performance, which is analogous to synaptic plasticity in the brain. Neurons receive input from other neurons through dendrites, process the information, and if the input is strong enough, they “fire” an electrical signal (action potential) through the axon to communicate with other neurons. Similarly to the brain, artificial neurons receive inputs from other nodes, compute a weighted sum of the inputs, and pass the result through an activation function (such as a sigmoid) to determine whether they “fire” (produce an output).

Convolutional neural networks (CNNs) are the most widely recognized biomimetic model for image processing, directly inspired by the visual cortex. The layers of a CNN function similarly to the layers of neurons in the brain, with early layers identifying simple image features and deeper layers recognizing more complex structures. In the human brain, visual processing occurs in a hierarchical manner through the visual cortex. Simple features like edges and orientations are detected in the early stages (e.g., V1 in the brain), while more complex features like shapes and object recognition occur in later stages (e.g., V4 and the inferotemporal cortex).

Explainable artificial intelligence (XAI) refers to AI models that not only perform tasks but also provide clear, understandable justifications for their decisions. XAI aims to address the black-box nature of many machine learning algorithms, such as deep neural networks, where decision processes are not easily interpretable by humans. Interestingly, there is a growing body of research exploring the parallels between explainable AI and the mechanisms of human cognition, especially regarding how the brain processes and explains its own decisions.

The human brain has the remarkable ability to process complex information and simultaneously generate explanations for the decisions it makes. XAI attempts to mimic this cognitive function by producing human-understandable reasons for AI decisions.

Both the human brain and many explainable AI systems rely on hierarchical processing to arrive at conclusions, breaking down complex problems into smaller, understandable components. The brain organizes information hierarchically, such as how sensory data are processed from lower-order areas (simple feature detection, like color or shape) to higher-order areas (complex scene interpretation or recognition). This layered organization helps us understand cause-and-effect relationships and explain decisions. Humans are naturally skilled at causal reasoning—the ability to explain outcomes based on cause-and-effect relationships. Our proposed method based on Bayesian networks attempts to instill similar capabilities in AI systems by allowing them not just to predict but also to explain outcomes based on causal models.

One of the key motivations behind XAI is to increase the trust between humans and AI systems. Likewise, in human cognition, self-explanation is critical for trust and collaboration among people. Regarding error detection and bias mitigation, both human cognition and XAI can detect and address biases or errors in decision-making processes.

One of the strongest links between XAI and the human brain concerns the explanatory/mental models. Humans form mental models—cognitive representations of how things work in the real world—to make sense of complex systems. XAI and especially BNs aim to align AI models with these human mental models, ensuring that AI decisions are consistent with human reasoning.

Thus, the connection between explainable AI and human brain functions is deep and multifaceted. Both aim to break down complex information into understandable units, provide reasoning for decisions, and align these decisions with a broader conceptual framework. As XAI continues to develop, drawing inspiration from cognitive neuroscience can lead to more intuitive, human-like explanations in AI systems. Likewise, insights from AI research may help us better understand human cognition, especially in the areas of decision making (diagnosis is a special case of decision making), attention, and learning.

As we explain in more detail in the following sections, our fully interpretable XAI method applies special algorithms that segment and identify the critical parts of an image using trained CNN. Subsequently, we construct a mental-like model with the help of a BN that encapsulates factors/concepts extracted from the image.

## 2. Related Work

While numerous studies have leveraged ML methods involving mammograms, CT images, and ultrasounds, there has been a notable lack of focus on ML methods with thermograms. It is known that thermograms offer critical health information that, when properly analyzed, can significantly aid accurate pathology determination and diagnoses.

Specialized NNs, trained on extensive databases, can systematically process medical images, incorporating patients’ complete medical histories to generate over 90% accurate diagnostic outcomes. Recent comprehensive reviews [21,25,26,27,30,31] on breast cancer detection using thermography have highlighted advantages and drawbacks of screening methods, the potential of thermography in this domain, and advancements in AI applications for diagnosing breast cancer.

Research [32] aimed to comprehensively evaluate the predictive distribution of all classes using such classifiers as naïve Bayes (NB), Bayesian networks (BNs), and tree-augmented naïve Bayes (TAN). These classifiers were tested on three datasets: breast cancer, breast cancer Wisconsin, and breast tissue. Results indicated that the Bayesian network (BN) algorithm achieved the best performance with an accuracy of 97.281%.

Nicandro et al. [33] explored BN classifiers in diagnosing breast cancer using a combination of image-derived and physician-provided variables. Their study, with a dataset of 98 cases, showcased BNs’ accuracy, sensitivity, and specificity, with different classifiers presenting varying strengths.

Ekicia et al. [34] also proposed software for automatic breast cancer detection from thermograms, integrating CNN and Bayesian algorithms. Their study, conducted on a database of 140 women, achieved a remarkable 98.95% accuracy.

A study by the authors [35] utilized CNN to execute and demonstrate a deep learning model based on thermograms. The presented algorithm was able to classify images as “Sick, Tumor” or “Healthy, No Tumor”. The main result of the study was an absence of image pre-processing by utilizing a data augmentation technique to artificially increase the dataset size, helping the CNN to learn and differentiate better in binary classification tasks.

The authors also conducted research on the integration of BN and CNN models [36], which showed that integration of the two methods can improve the results of systems that are based only on one of the models. Such an expert system is beneficial as it offers explainability, allowing physicians to understand the key factors influencing diagnostic decisions.

Further research has investigated several transfer learning models. It was shown that the transfer learning approach led to better results compared with the baseline approach. Among the four transfer learning models, MobileNet achieved one of the best results, with 93.8% accuracy according to most metrics; moreover, it demonstrated competitive results even when compared with state-of-the-art transfer learning models [37].

The current paper reports further development of our methodology to achieve interpretable diagnosis with the use of XAI algorithms that enhance the integration of BNs with CNNs, for detection of breast cancer at early stages.

## 3. Methodology

### 3.1. Overview

Developing an accurate medical expert model for diagnosing conditions requires substantial patient data. The dataset should encompass images as well as a variety of relevant factors, i.e., age, weight, and other medical history data. Using such a comprehensive dataset, our integrated CNN and BN network analyzes images and the relationships between these factors, expressed as conditional probabilities. Because ANNs and BNs operate based on statistical principles, the accuracy of their predictions relies heavily on the size of the dataset. Therefore, this research process involved several stages:Initial acquisition and preprocessing of data;Compilation of the collected data;Deployment and utilization of CNN+BN algorithm.

A methodology was developed, with key overall steps as follows: (1) thermal images as well as medical history data are collected; (2) segmentation of thermal images is performed; (3) the CNN is trained with the thermal images and makes a diagnosis; (4) XAI algorithms identify which are the critical parts of images; (5) statistical and computational factors are evaluated from the critical parts of the thermal images; (6) the BN is trained with the factors and the medical records dataset and makes a diagnosis; (5) if the BN has similar accuracy to the CNN, the structure of the BN reveals a full, interpretable model of the diagnosis decision; (6) Finally, in the BN we include as an extra factor, the diagnosis of the CNN. Running it again generates a very accurate expert system for diagnosis.

The described methodology is discussed in detail in the following sections/subsections.

### 3.2. Initial Data Collection

#### 3.2.1. Patients’ Medical Information

The presented research received certification from the Institutional Research Ethics Committee (IREC) of Nazarbayev University (identification number 294/17062020). Before undergoing thermographic imaging, patients were subjected to an introductory procedure and were required to provide their consent to participate in the research. Additionally, they were asked to complete questionnaires to provide us with medical information.

#### 3.2.2. Thermal Images

Thermal images were obtained from two separate sources to form a combined dataset. The first dataset was obtained from a publicly available breast research database [38] managed by international researchers. The second dataset of thermal images, collected locally as part of an ongoing project at Astana Multifunctional Medical Center [39], was obtained with the approval and certification of the Institutional Research Ethics Committee (IREC) of Nazarbayev University (identification number 294/17062020) and the hospital itself.

Local thermographic images (Figure 1) were captured using IRTIS 2000ME (10MED, Moscow, Russian) and FLUKE TiS60+ (Fluke, Everett, WA, USA) thermal cameras (Figure 2). These infrared cameras were employed to measure the temperature distribution on the surface of the breast. Healthy breast tissue usually exhibits uniform temperature distribution, while areas affected by cancer may experience disturbances such as increased temperature due to increased blood flow near the tumor and metabolic activity.

However, in thermograms, temperature points external to the body may influence the calculations. Therefore, the key steps of our proposed method are depicted in Figure 3. In the proposed method, thermograms are first preprocessed and the region of interest is then segmented.

Temperature readings are extracted from the thermographic images and saved in Excel format, selected for ease of calculation (Figure 4). Within this Excel file, the recorded temperatures correspond to distinct pixels within the thermographic image. Notably, the red regions within the image represent areas exhibiting higher temperatures relative to the surrounding regions. It is worth mentioning that the maximum temperature for the red color was adjusted within the program to ensure accurate analysis.

### 3.3. Segmentation

In this research, Python code was designed to systematically navigate Excel files and eliminate cells with temperatures below a specified threshold (for example, 29 °C). Figure 5 and Figure 6 show the Python code of the proposed model and its corresponding outcome, respectively.

### 3.4. Convolutional Neural Network Model

The CNN model utilizes thermal images in JPEG format as input data and produces binary output (1 for positive, 0 for negative), as elaborated in [35,36].

CNNs process data reminiscent of the grid processing seen in the LeNet architecture. In accordance with [35,36], the CNN consists of five data processing layers for alignment and two output layers (Figure 7). During the training phase, the CNN adjusts its parameters based on the provided dataset, gradually increasing its accuracy [35,36,37,38,39]. Transfer learning was used to adapt an existing model to the current task, speeding up the learning process (Figure 8). The dataset was divided into separate subsets for training, cross-validation, and testing.

To evaluate the model’s performance, including accuracy and precision, eight metrics were used. The confusion matrix was useful for identifying areas where the model made errors.

### 3.5. Explainable Artificial Intelligence (XAI) Framework

Explainable artificial intelligence (XAI) is a subset of machine learning (ML) focused on elucidating the processes by which ML models generate their outputs. Applying XAI to an ML model enhances its efficiency, as the reasoning behind the model’s inferences becomes traceable [40].

Artificial neural networks (ANNs) usually comprise numerous layers linked through complex, nonlinear relationships. Even if all these layers and their interconnections were examined, it would be nearly impossible to fully understand how the ANN arrived at its decision. Consequently, deep learning is frequently regarded as a ‘black box.’ Given the high stakes involved in medical decision making, it is unsurprising that medical professionals have expressed concerns about the opaque nature of deep learning, which currently represents the leading technology in medical image analysis. As a consequence, novel XAI methods have been developed trying to provide interpretable deep learning solutions [40,41,42,43,44,45,46,47,48].

LIME, which stands for local interpretable model-agnostic explanations, is an algorithm designed to faithfully explain the predictions of any classifier or regressor by locally approximating it with an interpretable model. LIME offers local explanations by substituting a complex model with simpler ones in specific regions. For instance, it can approximate a CNN with a linear model by altering the input data, and then, the output of the complex model changes. LIME uses the simpler model to map the relationship between the modified input data and the output changes. The similarity of the altered input to the original input is used as a weight, ensuring that explanations from the simple models with significantly altered inputs have less influence on the final explanation [47,48].

First of all, the digit classifier was built by installing TensorFlow, specifically employing Keras, which is installed in TensorFlow. The Keras frontend simplifies the complexity of lower-level training processes, making it an excellent tool for quickly building models.

Keras includes the MNIST dataset in its distribution, which can be accessed using the load_data() method from the MNIST module. This method provides two tuples containing the training and testing data organized for supervised learning. In the code snippet, ‘x’ and ‘y’ are used to represent the images and their corresponding target labels, respectively.

The images returned by the method are 1-D numpy arrays, each with a size of 784. These images are converted from unit8 to float32 and reshaped into 2-D matrices of size 28 × 28. Since the images are grayscale, their pixel values range from 0 to 255. To simplify the training process, the pixel values are normalized by dividing by 255.0, which scales them between 0 and 1. This normalization step is crucial because large values can complicate the training process.

A basic CNN model was developed to process 3-D images by passing them through Conv2D layers with 16 filters, each sized 3 × 3 and using the ReLU activation function. These layers learn the weights and biases of the convolution filters, essentially functioning as the “eyes” of the model to generate feature maps. These feature maps are then sent to the MaxPooling2D layer, which uses a default 2 × 2 max filter to reduce the dimensionality of the feature maps while preserving important features to some extent.

The basic CNN model was trained for 2 epochs with a batch size of 32 through model.fit() and a validation set, which was set aside earlier while loading MNIST data, is used. In this context, “epochs” denotes the total number of times the model sees the entire training data, whereas “batch size” refers to the number of records processed to compute the loss for one forward and backward training iteration.

With the model ready, LIME for explainable AI (XAI) could be applied. The lime image module from the LIME package was employed to create a LimeImageExplainer object. This object has an explain_instance() method that takes 3-D image data and a predictor function, such as model.predict, and provides an explanation based on the predictions made by the function.

The explanation object features a get_image_and_mask() method, which, given the predicted labels for the 3-D image data, returns a tuple of (image, mask). Here, the image is a 3-D numpy array and the mask is a 2-D numpy array that can be used with skimage.segmentation.mark_boundaries to show the features in the image that influenced the prediction.

The results of the applied algorithm are more thoroughly presented and discussed in the Results and Discussion section of the paper.

### 3.6. Informational Nodes for the Diagnosis

Following the XAI algorithms, the critical region of interest (ROI) (each breast separately) is isolated from the image files. At this stage, temperature data of both healthy and tumor-affected breasts, stored in the spreadsheet format, are used to calculate various statistical/computational parameters. Each temperature value within the spreadsheet cell corresponds to a singular pixel from the thermal image. It is worth noting that based on available medical information, a determination is made between a healthy and an affected breast. In cases where there is no tumor, for calculations, the left breast is considered affected, and the right breast is considered healthy. Temperature values are measured in degrees Celsius. Below is a complete list of parameters used for these calculations.

Maximum TemperatureMinimum TemperatureTemperature Range (Maximum minus Minimum Temperature)MeanMedianStandard DeviationVarianceDeviation from the Mean (Maximum minus Mean Temperature)Deviation from the Maximum Temperature of the Healthy Breast (Maximum minus Maximum Temperature of the Healthy Breast)Deviation from the Minimum Temperature of the Healthy Breast (Maximum minus Minimum Temperature of the Healthy Breast)Deviation from the Mean Temperature of the Healthy Breast (Maximum minus Mean Temperature of the Healthy Breast)Deviation between Mean Temperatures (Mean minus Mean Temperature of the Healthy Breast)Distance between Points of Maximum and Minimum Temperature:
$$D=\sqrt{(Number\;of\;pixels\;left\;or\;right{)}^{2}+{\left(Number\;of\;pixels\;up\;or\;down\right)}^{2}}$$A = Number of Pixels near the Maximum with Temperature Greater than
[mean+0.5×(maximum−mean)]B = Number of Pixels of the Entire AreaC = Number of All Pixels with Temperature Greater than
[mean+0.5×(maximum−mean)]Ratio of A to B (A/B)Ratio of C to B (C/B)

The calculated factors, along with the initially gathered patient information, are merged into a unified file format compatible with the software requirements.

### 3.7. Bayesian Network (BN) Model

A Bayesian network (BN), also known as a belief network or probabilistic directed acyclic graphical model, is defined mathematically as a graph structure and more specifically as a directed acyclic graph (DAG), **G** = (V, E) where *V* is a set of vertices (nodes) representing random variables and E is a set of directed edges (arcs) representing conditional dependencies between the variables. Each node $X_{i}$ in the network is associated with a conditional probability distribution $P(X_{i}|\{Parents\}(X_{I}))$, where the parent nodes of $X_{i}$ are determined by the topology of the graph **G**.

The main property of BN is a theorem that simplifies calculation of the joint probability distribution. Let $X=\{X_{1},\;X_{2},\ldots ,X_{n}\}$ be a set of ***n*** random variables represented by the nodes in the DAG, **G**. Then, the joint probability $\left(X_{1},X_{2},\ldots ,X_{n}\right)$ of the set of random variables X can be factorized as:$$P(X_{1},\;X_{2},\ldots ,X_{n})=\prod_{i=1}^{n}P(X_{1}|\{Parents\}(X_{i}))$$

Furthermore, each node $X_{i}$ is conditionally independent of its non-descendants, given its parents. This is expressed as:$$P(X_{i}|\{Non\_Descendants\}(X_{i}),\;\{Parents\}(X_{i}))=P(X_{i}|\{Parents\}(X_{i}))$$

Thus, BNs provide an efficient way to represent the joint probability distribution by exploiting conditional independencies. Inference in BNs involves computing the posterior distribution of a set of query variables, given evidence about some other variables. By defining the structure and the conditional probability distributions for each node, we can model complex probabilistic relationships using BNs.

The BN models that we have constructed include as informational nodes all previously mentioned factors presented in Section 3.6 as well as historical medical record data. Finally, as an additional factor, we can also include in the BN, the diagnosis from the initial CNN model.

The diagnosis variable was integrated into the compiled data file, where a value of 1 signified a positive diagnosis and 0 denoted a negative one. The file was then inputted into the software (BayesiaLab 11.2.1 [49]), categorizing different parameters as either continuous or discrete (Figure 9).

The final diagnosis decision was specifically designated as a target variable for the whole BN. Subsequently, the software computed the frequencies between the provided parameters and the target variable, presenting the outcomes as conditional probabilities. Using this, the connections between nodes were established during training, finalizing the acyclic graph (DAG) (Figure 10).

Supervised learning, including augmented naïve Bayes, was used to obtain results, and our findings were validated using K-fold analysis. In BayesiaLab, K-fold cross validation is a method used to evaluate the performance of Bayesian network models. It involves dividing the dataset into K subsets (or folds), where one subset is used as the test set and the remaining K-1 subsets are used to train the model.

## 4. Results and Discussion

### 4.1. Data Compilation Results

The compiled dataset consisted of 364 images, of which 153 thermal images were categorized as “sick” and 211 thermal images were categorized as “healthy” according to the physician’s diagnosis (Table 2).

Images from a total of 364 thermograms were further processed with XAI algorithms. The factors together with the patients’ medical history data were prepared for integration into BayesiaLab 11.2.1 [49].

We constructed two models. Model A was able to discover all influential factors that drove the diagnosis, see Figure 10. It provided full explainability to our expert model. Model B is the final expert model, giving the best possible diagnosis.

### 4.2. Results of Convolutional Neural Network with XAI

In this case, XAI depicted areas that contributed most when classifying images and assigning them probabilities of being classed “Sick” or “Healthy”. The LIME library that was used to analyze images and the CNN models highlighted areas of thermal images of breast that clearly showed areas indicating movement in the development of cancerous tissues. Explainable parts of images are shown in Figure 11 with their respective diagnoses and classification results.

The XAI algorithm identified the critical ROI, meaning the part of thermogram critical to the decision making, in most of the sick cases, see Figure 11. Furthermore, in most of the healthy cases, the XAI did not identify one particular region of the thermogram, see Figure 11.

### 4.3. Bayesian Network Results—Model A

Model A integrates medical records information and the factors from the thermal images that have been evaluated from critical parts of images with the help of XAI algorithm. Model B also includes diagnostic data from a CNN model.

The combined dataset was used to train the network and evaluate its performance. Results of both models were developed and analyzed using BayesiaLab 11.2.1 software, specifically using a supervised learning approach, tree-augmented naïve Bayes. Additionally, the K-fold verification method did not reveal any noticeable shortcomings of the BN models, and this indicates their efficiency.

As illustrated in Figure 12, the nodes showing different influences were assessed using mutual information assessment. The model displayed robustness and efficiency with a dataset of 364 patients, resulting in high accuracy.

The influential factors identified were in line with the processed data from both unsupervised and supervised learning, as shown in Table 3.

An important measure from Table 4 is the Gini index, which reflects the equilibrium between positive and negative values of the target variable across the dataset. This equilibrium is crucial for the model to discern between the two conditions successfully and make precise predictions in both instances. An ideal Gini index, representing perfect balance, is set at 50%. In our case, the value registered at 43.04895%, which is relatively proximate to the optimal equilibrium.

Table 5 shows very good results for the XAI expert model, particularly highlighting the rates of true positives (79.5%) and true negatives (87.68%). Only 58 patients out of 364 were misclassified by the model. The model performance shown in Figure 13 below is depicted using a receiver operating characteristic (ROC) curve. The overall accuracy of the model stands at 84.07%, as shown in Table 6.

### 4.4. Bayesian Network with Convolutional Neural Network Results—Model B

Model B was similar to the previous model but also integrated predictions from a CNN model. The final directed acyclic graph (DAG) is presented in Figure 14. Including additional information improved performance and strengthened the model’s robustness. The CNN model effectively differentiated between healthy and diseased samples, achieving an accuracy rate of 88%. Following this classification, the outcomes were integrated into the dataset for Model B.

The influential factors identified were in line with data using both unsupervised and supervised learning, as shown in Table 7.

The results are reported in Table 7, Table 8, Table 9 and Table 10 and Figure 13. Table 10 illustrates the significant improvement in model performance after integrating the CNN variable, resulting in fewer incorrect predictions. Of the 364 cases, only 33 were inaccurately diagnosed, resulting in an overall accuracy of 90.9341%. The Gini index increased to 46.9272% due to the addition of the CNN node. In addition, the ROC index increased from 93.049% to 96.9272%.

### 4.5. Comparative Assessment

This section compares the performance of Model B with previous research studies that have also used thermal imaging. Data were collected from various research papers reporting the use of thermal imaging technology, to obtain an idea of how Model B compared. This analysis aimed to understand how well Model B performed compared with existing approaches using thermal imaging. Table 11 shows a summary of the results.

The table of comparative analysis presents the performance of Model B within the context of studies focused on thermal imaging. Notably, Model B exhibited competitive accuracy with a performance of 90.9341%, closely matching the results of several previous studies. It is worth noting that some of these previous studies may have used smaller datasets than ours, which could potentially have affected the accuracy of their results. Moreover, our use of XAI integration of CNNs and BNs provides a clear advantage.

Model B in the present study had somewhat lower accuracy compared with Model B of reference [37]. The reason is that we have enlarged the dataset by including different sets of images with some of them missing medical records. However, the aim of this paper is to demonstrate the efficiency of the proposed XAI methodology. It is well known that one can always achieve higher accuracy with better training based on larger, high-quality datasets.

These comparisons provide valuable information to both researchers and practitioners, guiding future developments in thermal imaging applications and highlighting ways to optimize performance of XAI expert models in real-world scenarios.

## 5. Conclusions

Breast cancer continues to represent a major global health problem, highlighting the need for advanced diagnostic techniques to improve survival rates through early detection. The literature highlights the growing burden of breast cancer and the limitations of current diagnostic methods, stimulating the study of innovative approaches. For detection of breast-cancer, thermography has emerged as a cost-effective and non-invasive alternative. This study extensively explored machine learning techniques, specifically Bayesian networks combined with convolutional neural networks, to improve interpretable early detection of breast cancer.

Explainable artificial intelligence offers transparent and interpretable decision-making processes, promoting trust and collaboration between humans and artificial intelligence systems in several application domains. In the present study, we used explainable artificial intelligence algorithms to identify the critical parts of thermal images and subsequently, we integrated convolutional neural networking and Bayesian networking to build a highly accurate expert model for diagnosis.

The methodology included diverse datasets, facilitating the development of robust Bayesian network models for accurate possible breast cancer diagnosis. Using thermal images and medical records, Model A achieved an accuracy of 84.07%, while Model B, combining convolutional neural network predictions, achieved a 90.9341% accuracy. Model A had a similar performance with a sole convolutional neural networks model and therefore, we can trust it as a practically acceptable model that is fully readable by a physician. From Model A, we can easily read the interpretation of the diagnosis. We can observe which factors are dominant as well as their interactions. These results highlight the potential of interpretable machine learning methods to improve possible breast cancer diagnosis, with high accuracy and reducing the risk of misdiagnosis.

Moreover, the study models demonstrated competitiveness with other peer-reviewed literature articles, confirming their effectiveness and relevance. Both models performed well despite the relatively small dataset size, as confirmed by K-fold cross-validation. Future research will focus on increasing the size of the network training dataset and improving the Gini index to ensure a more balanced dataset and thereby improve performance.

In conclusion, this study contributes to advancing interpretable machine learning in healthcare by introducing an innovative approach to breast cancer detection. Successful integration of thermography, convolutional neural networks, and Bayesian networks promises to improve outcomes and quality of patient care in the fight against breast cancer. Given that thermography has been cleared by the FDA only as an adjunctive tool, meaning it should only be used alongside a primary diagnostic test like mammography, not as a standalone screening or diagnostic tool, continued research and validation are thus critical to exploit these methodologies’ potential.

## Figures and Tables

**Figure 1 biomimetics-09-00609-f001:**
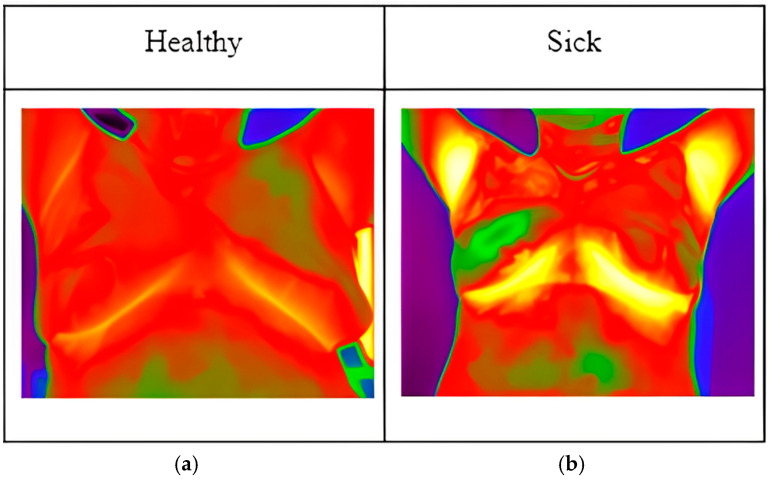
Thermographic images: (**a**) image of healthy patient; (**b**) image of sick patient.

**Figure 2 biomimetics-09-00609-f002:**
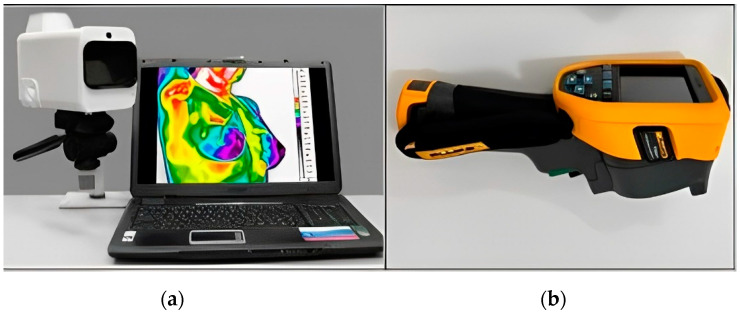
Thermal imaging cameras: (**a**) IRTIS 2000ME; (**b**) FLUKE TiS60+.

**Figure 3 biomimetics-09-00609-f003:**
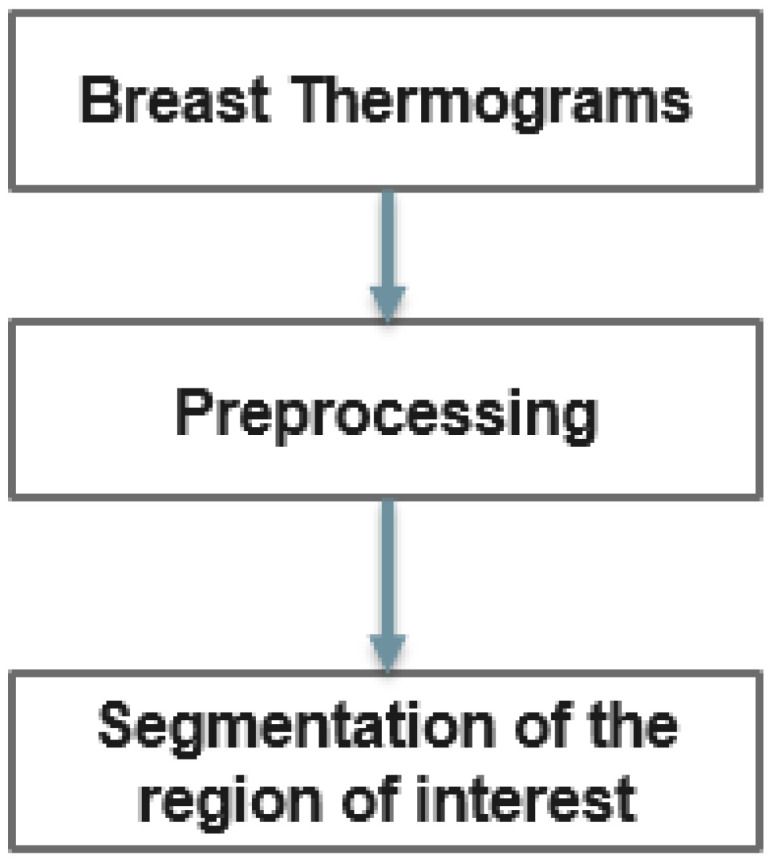
Diagram for detecting breast abnormalities using thermography.

**Figure 4 biomimetics-09-00609-f004:**
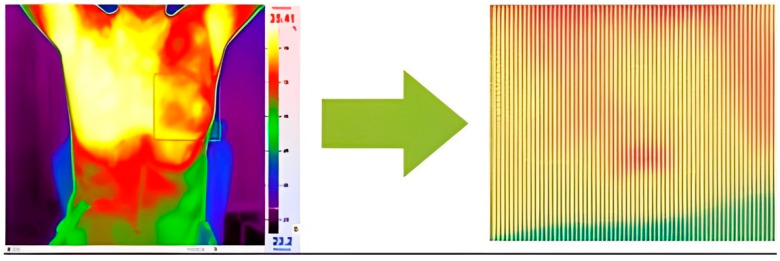
Transforming thermal images into an array of temperature values per pixel.

**Figure 5 biomimetics-09-00609-f005:**
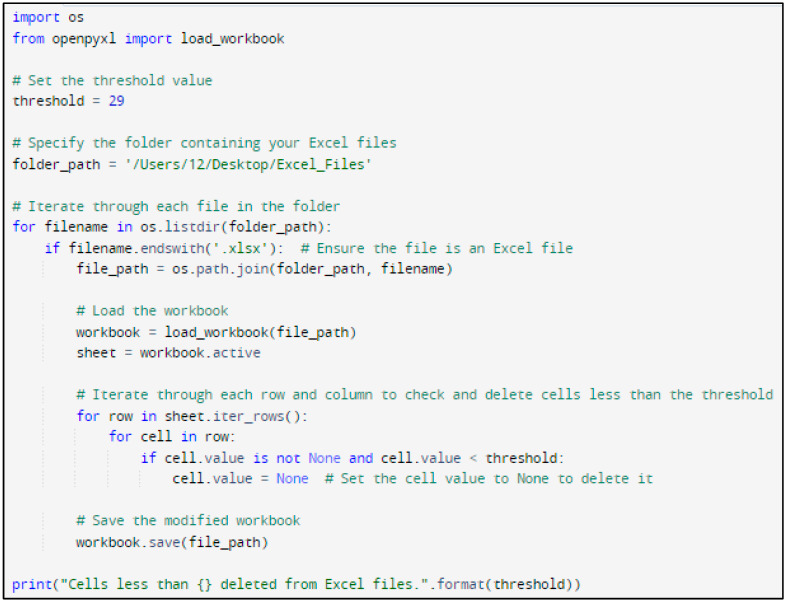
Python code of the proposed model.

**Figure 6 biomimetics-09-00609-f006:**
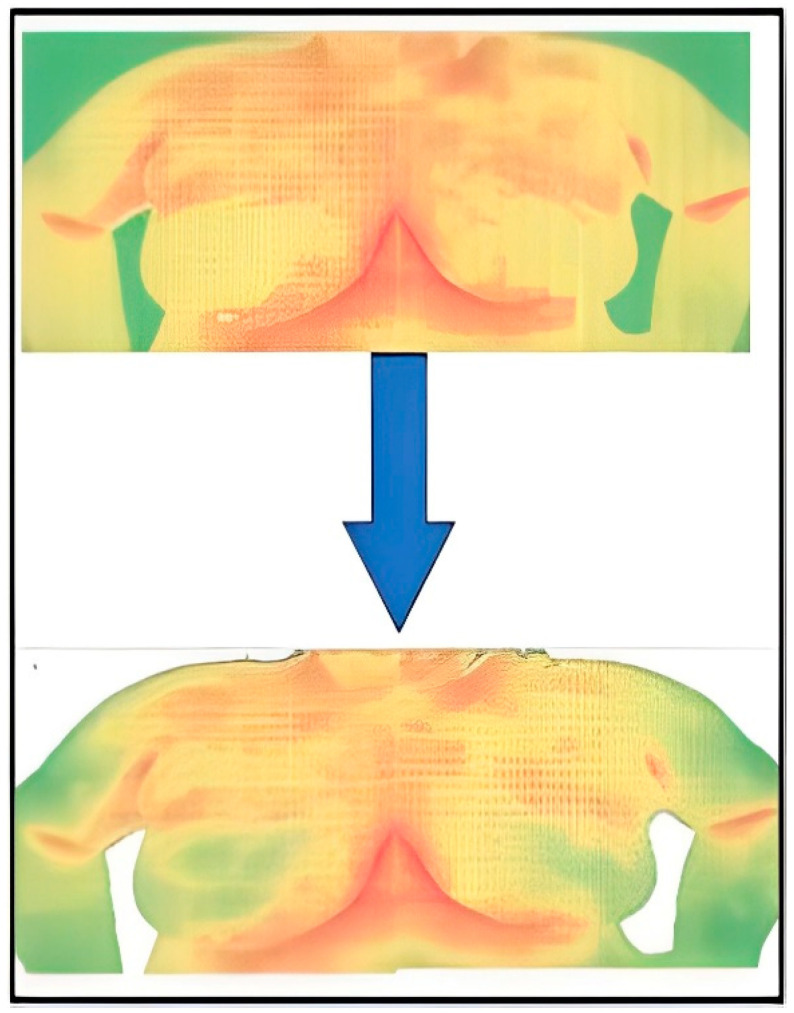
Result of the segmentation.

**Figure 7 biomimetics-09-00609-f007:**
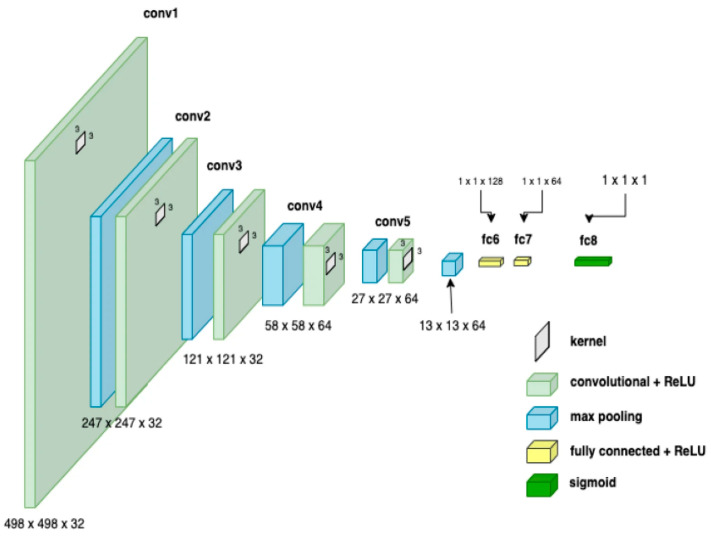
CNN architecture showing parameters at each level.

**Figure 8 biomimetics-09-00609-f008:**
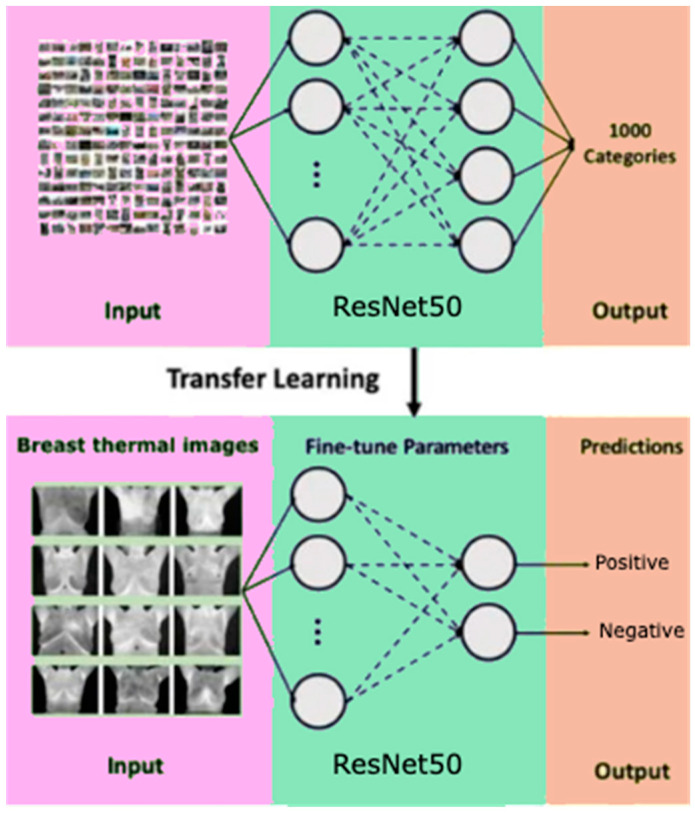
Scheme of transfer learning method for binary classification.

**Figure 9 biomimetics-09-00609-f009:**
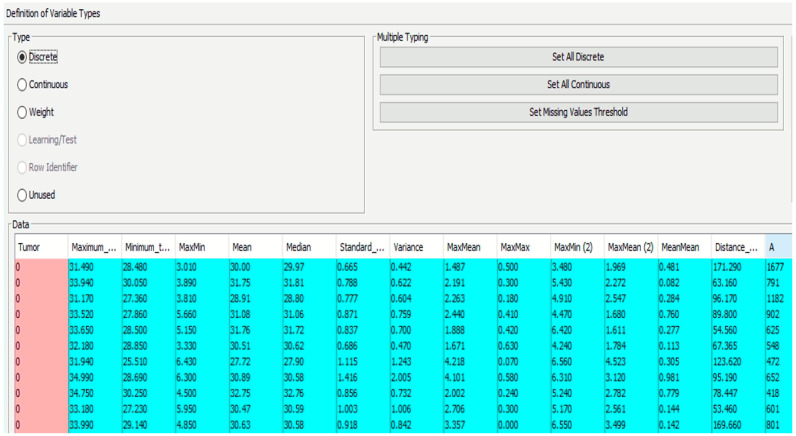
Definition of variables (continuous/discrete).

**Figure 10 biomimetics-09-00609-f010:**
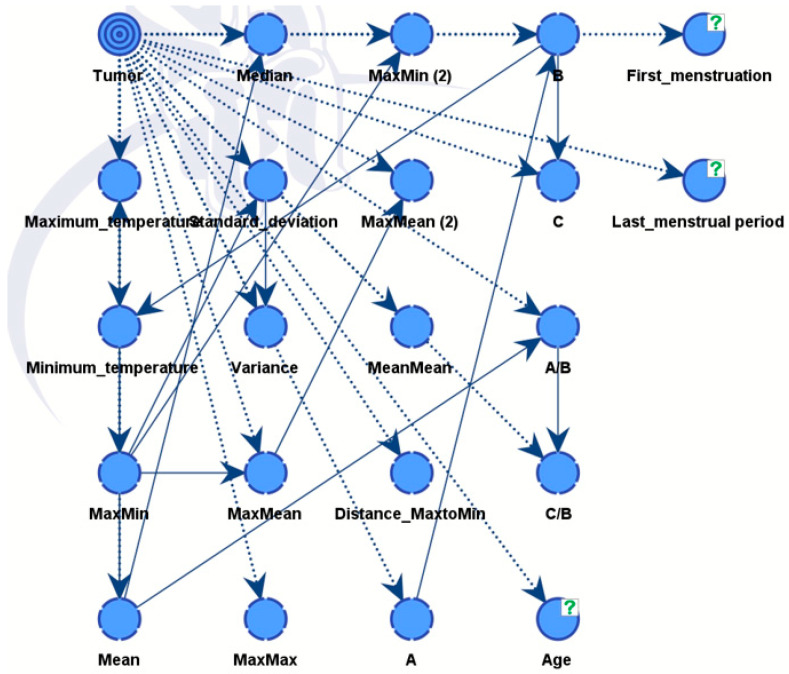
The link (relationship) between nodes through training in DAG.

**Figure 11 biomimetics-09-00609-f011:**
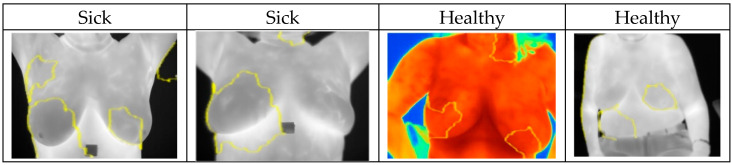
Highlighted areas of XAI library images showing parts that contributed to the CNN decision making.

**Figure 12 biomimetics-09-00609-f012:**
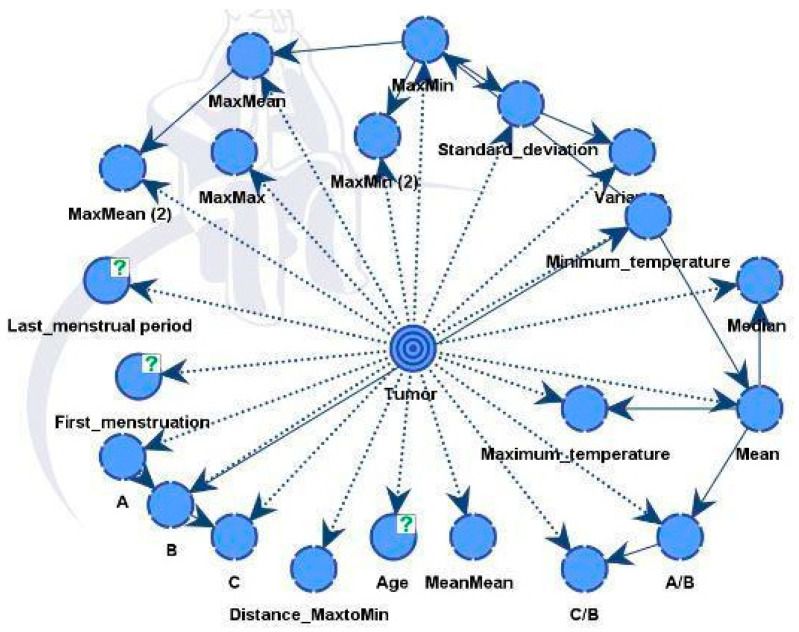
Model A and Factors Influencing the Tumor Target Variable.

**Figure 13 biomimetics-09-00609-f013:**
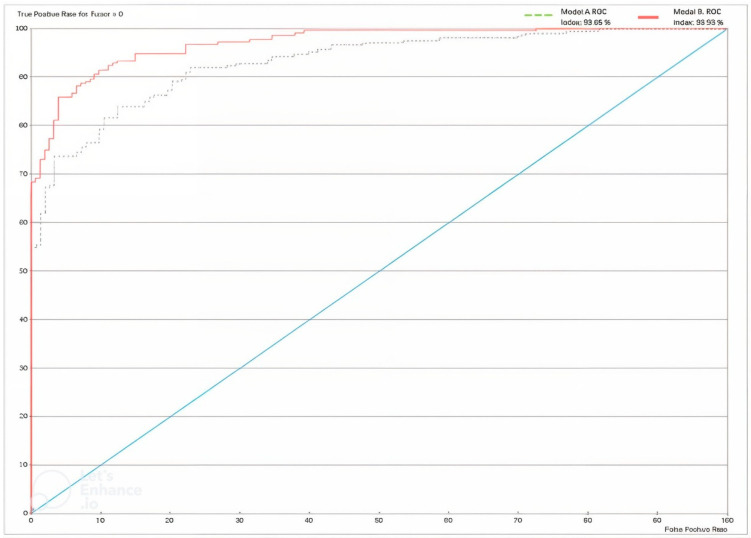
ROC Curve for Model A and Model B. For definitions, see https://www.bayesia.com/bayesialab/user-guide. Accessed on 1 September 2024.

**Figure 14 biomimetics-09-00609-f014:**
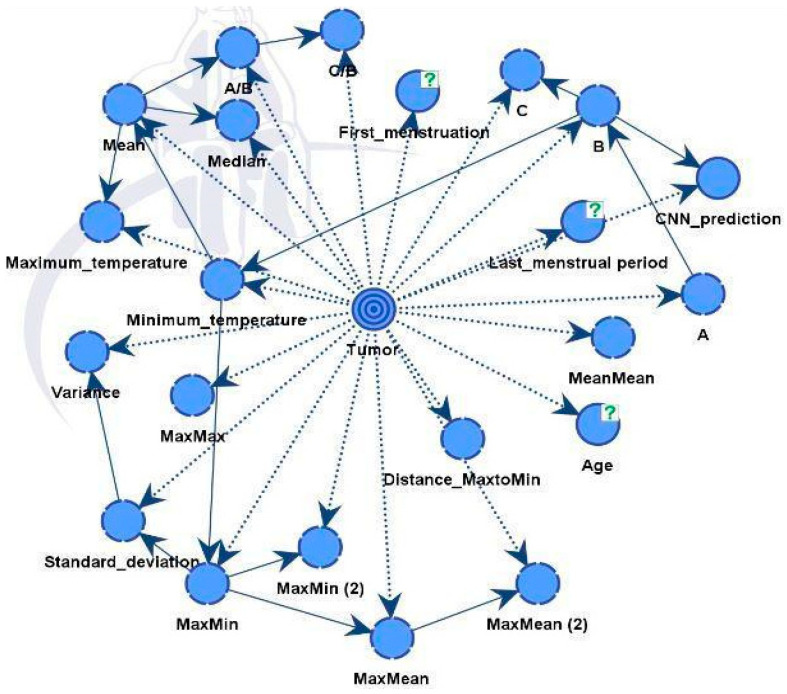
Model B and factors influencing the tumor target variable.

**Table 1 biomimetics-09-00609-t001:** Evaluation of current breast cancer detection methodologies [15].

Methods	Radiation	Sensitivity	Tumor Size	Advantage	Disadvantage
Mammography	X-rays	84%	≤2 cm	Simultaneous screening of bone, soft tissues, and blood vessels on a mass scale.	Preferentially chosen for individuals above 40 years old due to considerations regarding the impact of ionizing radiation.
Ultrasound	Sound waves	82%	2 cm	Affordable evaluation of breast lumps, with safety considerations.	Low resolution.
Magnetic resonance imaging (MRI)	RF waves	95%	≤2 cm	Non-invasive and safe.	The reconstruction process encounters an ill-posed problem.
Thermography	No radiation	85%	1 cm	Cost-effective, non-invasive, and safe.	Limited in its ability to detect tumors located deeper within the body.
Computer tomography (CT)	Ionizing radiation	81.2%	1 cm	Non-invasive, powerful, creates clear images on the computer, does not depend on density of the breast.	X-ray exposure, expensive, limited accessibility.

**Table 2 biomimetics-09-00609-t002:** Information about the dataset.

	Healthy	Sick	Total
Information gathered from a publicly accessible database [38]	166	100	266
Astana Medical Center [39]	45	53	98
Total	211	153	364

**Table 3 biomimetics-09-00609-t003:** Influence on tumor in model A. For definitions, see https://www.bayesia.com/bayesialab/user-guide. Accessed on 1 September 2024.

Node	Mutual Information	Normalized Mutual Information	Relative Mutual Information	Relative Significance	Prior Mean Value	G-Test	df	*p*-Value	G-Test (Data)	df (Data)	*p*-Value (Data)
Max_Temperature	0.2648	26.4847%	26.9807%	1.0000	34.0704	133.6448	2	0.0000%	133.7700	2	0.0000%
Mean	0.2173	21.7323%	22.1393%	0.8206	31.5696	109.6638	2	0.0000%	109.7311	2	0.0000%
Median	0.2135	21.3515%	21.7514%	0.8062	31.5531	107.7421	2	0.0000%	107.8073	2	0.0000%
Min_temperature	0.1881	18.8055%	19.1577%	0.7101	29.0253	94.8947	2	0.0000%	94.9764	2	0.0000%
Age	0.1860	18.6007%	18.9491%	0.7023	57.0016	93.8614	66	1.3715%	94.4754	66	1.2300%
Last_menstrual period	0.1453	14.5310%	14.8032%	0.5487	48.1478	73.3252	38	0.0503%	87.4606	38	0.0009%
B	0.0522	5.2226%	5.3204%	0.1972	28,337.8891	26.3539	2	0.0002%	26.5726	2	0.0002%
MaxMax	0.0450	4.4994%	4.5837%	0.1699	0.4621	22.7045	2	0.0012%	22.8927	2	0.0011%
MaxMin	0.0314	3.1357%	3.1944%	0.1184	5.0454	15.8229	2	0.0367%	15.8454	2	0.0362%
A/B	0.0223	2.2306%	2.2724%	0.0842	0.0369	11.2559	2	0.3596%	11.2609	2	0.3587%

**Table 4 biomimetics-09-00609-t004:** Performance Summary of Model A. For definitions see, https://www.bayesia.com/bayesialab/user-guide. Accessed on 1 September 2024.

Target: Tumor	
Gini Index	43.04895%
Relative Gini Index	86.0979%
Lift index	1.62245
Relative Lift Index	95.31725%
ROC Index	93.049%
Calibration Index	89.19985%

**Table 5 biomimetics-09-00609-t005:** Model A Confusion Table. For definitions, see https://www.bayesia.com/bayesialab/user-guide. Accessed on 1 September 2024.

Occurrences		
Value	0 (211)	1 (153)
0 (203)	178	25
1 (161)	33	128
Reliability		
Value	0 (211)	1 (153)
0 (203)	87.6847%	12.3153%
1 (161)	20.4969%	79.5031%
Precision		
Value	0 (211)	1 (153)
0 (203)	84.3602%	16.3399%
1 (161)	15.6398%	83.6601%

**Table 6 biomimetics-09-00609-t006:** Overall Accuracy of Model A. For definitions, see https://www.bayesia.com/bayesialab/user-guide. Accessed on 1 September 2024.

Classification Statistics	
Overall Precision	84.0659%
Mean Precision	84.0102%
Overall Reliability	84.2457%
Mean Reliability	83.5939%

**Table 7 biomimetics-09-00609-t007:** Influence on tumor in Model B.

Node	Mutual Information	Normalized Mutual Information	Relative Mutual Information	Relative Significance	Prior Mean Value	G-Test	df	*p*-Value	G-Test (Data)	df (Data)	*p*-Value (Data)
CNN prediction	0.4073	40.7339%	41.4967%	1.0000	0.3599	205.5477	1	0.0000%	205.6822	1	0.0000%
Min_temperature	0.2607	26.0710%	26.5592%	0.64000	29.0253	131.5569	2	0.0000%	131.6458	2	0.0000%
Max_temperature	0.2580	25.8026%	26.2859%	0.6334	34.0704	130.2030	2	0.0000%	130.3019	2	0.0000%
Mean	0.2251	22.5134%	22.9350%	0.5527	31.5695	113.6049	2	0.0000%	113.6820	2	0.0000%
Median	0.1999	19.9856%	20.3599%	0.4906	31.5530	100.8498	2	0.0000%	100.9124	2	0.0000%
Age	0.1860	18.6007%	18.9491%	0.4566	57.0016	93.8614	66	1.3715%	94.4754	66	1.2300%
Last_menstrual period	0.1453	14.5310%	14.8032%	0.3567	48.1478	73.3252	38	0.0503%	87.4606	38	0.0009%
MaxMin	0.0617	6.1739%	6.2895%	0.1516	5.0452	31.1542	2	0.0000%	31.1756	2	0.0000%
MaxMax	0.0423	4.2325%	4.3118%	0.1039	0.4620	21.3579	2	0.0023%	21.3938	2	0.0023%
A/B	0.0413	4.1279%	4.2052%	0.1013	0.0379	20.8299	2	0.0030%	22.7747	2	0.0011%
B	0.0383	3.8263%	3.8980%	0.0939	28,334.7862	19.3081	2	0.0064%	19.3214	2	0.0064%

**Table 8 biomimetics-09-00609-t008:** Performance Summary of Model B.

Target: Tumor	
Gini Index	46.9272%
Relative Gini Index	93.8544%
Lift index	1.67005
Relative Lift Index	98.02405%
ROC Index	96.9272%
Calibration Index	88.3618%

**Table 9 biomimetics-09-00609-t009:** Model B Confusion Table.

Occurrences		
Value	0 (211)	1 (153)
0 (203)	195	17
1 (161)	16	136
Reliability		
Value	0 (211)	1 (153)
0 (203)	91.9811%	8.0189%
1 (161)	10.5263%	89.4737%
Precision		
Value	0 (211)	1 (153)
0 (203)	92.4171%	11.1111%
1 (161)	7.5829%	88.8889%

**Table 10 biomimetics-09-00609-t010:** Overall Accuracy of Model B.

Classification Statistics	
Overall Precision	90.9341%
Mean Precision	90.6530%
Overall Reliability	90.9272%
Mean Reliability	90.7274%

**Table 11 biomimetics-09-00609-t011:** Model B results compared to previous studies.

#	Accuracy
Baffa et al. [50]	95%
Torres-Galván et al. [51]	92.3%
Zhang et al. [52]	93.83%
Farooq et al. [53]	80%
Nicandro et al. [33]	76.10%
Ekicia et al. [34]	98.95%
Model B	90.9341%

## Data Availability

Data are available at http://visual.ic.uff.br/dmi/, accessed on 1 January 2024 and https://sites.google.com/nu.edu.kz/bioengineering/dataset?authuser=0, accessed on 1 January 2024.

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
