# Peer review of "Fully Interpretable Deep Learning Model Using IR Thermal Images for Possible Breast Cancer Cases"

_biomimetics, 2024, doi:10.3390/biomimetics9100609_

Round 1

Reviewer 1 Report

Comments and Suggestions for Authors

The authors collected thermal imaging data of a cohort of 364 patients (153 sick, 211 healthy) and developed two models for early breast cancer detection: A) a Bayesian Networks model developed based on LIME-selected critical parts of images; and B) model A integrated with the results of a pre-developed CNN model. the model developed could potentially be applied to automate the labor-intensive evaluation task of evaluating thermal images for physicians.

Overall, the report is not well-organized. I have a few comments / concerns for further improvement of the report as outlined below.

1.        Line 157-159 is completely off and irrelevant to the paper.

2.        I am confused about the main contribution of this research. I understand that developing a computer-aided model can help replace the labor-intensive manual chart/imaging review process. However, to the best of my knowledge, thermography does not detect breast cancer; it merely alerts individuals to changes, indicating the need for further investigation. The authors stated that the developed model has the “potential to transform breast cancer diagnosis, improving accuracy and reducing the risk of misdiagnosis” in both Abstract and the Report Body. These statements should be revised.

3.        In the Introduction section, the authors stated that thermography was highlighted as a primary screening method for detecting breast cancer in recent studies. However, only one reference is provided (#20 Alshayeji et al.) and it does not provide evidence to support the claim.

4.        For Table 1, please also provide references for the listed screening methods. According to a paper published in 2014, the overall sensitivity of mammography is about 87%: Tosteson AN, Fryback DG, Hammond CS, et al. Consequences of false-positive screening mammograms. JAMA Intern Med. 174(6):954-61, 2014.

5.        This research is based a previously-published research by the same team (Aidossov, N., Zarikas, V., Zhao, Y., Mashekova, A., Ng, E. Y. K., Mukhmetov, O., Mirasbekov, Y., & Omirbayev, A. (2023). An Integrated Intelligent System for Breast Cancer Detection at Early Stages Using IR Images and Machine Learning Methods with Explainability. SN computer science4(2), 184. https://doi.org/10.1007/s42979-022-01536-9). In their previous paper, an integrated BN + CNN model was developed for the same purpose and achieved an overall precision of 92.1053%, which is slightly higher than the model B presented in this paper (90.9341%). Could the author clarify whether adding the XAI decreases the model performance?

6.        I am not sure about the point of developing model A.

Comments on the Quality of English Language

Moderate editing of English language required.

Reviewer 2 Report

Comments and Suggestions for Authors

Journal: Biomimetics:

Title: Further development and validation of an interpretable deep learning model for IR image based breast cancer diagnosis

Authors: Yerken Mirasbekov, Nurduman Aidossov, Aigerim Mashekova*, Vasilios Zarikas, Michael Yong Zhao, Eddie Yin Kwee Ng, Anna Midlenko

Cancer diagnosis is one of the most important tasks in medicinal chemistry nowadays.

The authors have applied two machine learning (ML) techniques: Bayesian Networks (BN) and Convolutional Neural Networks (CNN). Their combination is a suitable tool to achieve some betterment as compared to earlier models.

The authors’ claim was that [their “study models” have] “competitiveness with other peer reviewed literature articles, confirming their effectiveness and relevance”. However, the statement is bold and not entirely true.

The final comparison table consist of seven models including their Model B. For literature models excel better than the authors’ model B. This alone questions the usefulness of the new model. Why is to ‘further develop’ existing models with worse performance?

In any case the number of performance parameters (accuracy and ROC curve) is too small to achieve any reasonable conclusion. Recommended reading:

Anita Rácz, Dávid Bajusz*, Károly Héberger, Multi-Level Comparison of Machine Learning Classifiers and Their Performance Metrics, Molecules, 24, (2019) 2811

https://doi.org/10.3390/molecules24152811

Minor errors

Title

The title is not focused enough: there are superfluous words (Further, based, model, etc. Abbreviations should be avoided in titles abstract, highlights and conclusions (without resolving).

“Table 1. Evaluation of current breast cancer detection methodologies” – The source is missing.

Such important terms as “reliability” and “Lift index” have not been defined (and used loosely).

“Table 3. Influence on tumor in model A” What are the numbers in the table? What is B? What is G-test? A probability (p-value) cannot be larger than 1 (p=1.3715) ??? Relative mutual information. Relative to what? A table should be understandable on their own, with explanations and definitions as footnotes or referring to them in the text.

Table 11 results? Accuracies?

Figure 11 and the statement in the conclusion “XAI offers transparent and interpretable decision-making processes” contradict each other.

Etc. Etc.

In summary, the manuscript cannot be accepted in its present form. A resubmission is also questionable as the authors need to provide a detailed multicriteria decision analysis using considerably more performance parameters. However, the combination of Bayesian- and Convolutional Neural Networks will not provide better models, at all probability.

Sep.13 / 2024              referee:

Round 2

Reviewer 1 Report

Comments and Suggestions for Authors

The authors have addressed my comments.
